# Chromosome Engineering in Tropical Cash Crops

**Pablo Bolaños-Villegas** [1,2] 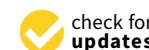

1   Fabio Baudrit Agricultural Research Station, University of Costa Rica, La Garita, Alajuela 20101, Costa Rica;
    pablo.bolanosvillegas@ucr.ac.cr
2   Jardín Botánico Lankester, Universidad de Costa Rica, Cartago P.O. Box 302-7050, Costa Rica

**Abstract:** Tropical and subtropical crops such as coffee, cacao, and papaya are valuable commodities, and their consumption is a seemingly indispensable part of the daily lives of billions of people worldwide. Conventional breeding of these crops is long, and yields are threatened by global warming. Traditional chromosome engineering and new synthetic biology methods could be used to engineer new chromosomes, facilitate the transmission of wild traits to improve resistance to stress and disease in these crops, and hopefully boost yields. This review gives an overview of these approaches. The adoption of these approaches may contribute to the resilience of agricultural communities, lead to economic growth and secure the availability of key resources for generations to come.

**Keywords:** tropical cash crops; coffee; cacao; papaya; chromosome engineering; synthetic biology

## 1. Introduction

By the year 2050, the world population may reach 9 billion and the demand for food may grow by 70% [1]. Unfortunately, climate change may increase the frequency of drought and intense precipitation events as well as elevate temperatures, which may exacerbate food insecurity and instability caused by low agricultural diversity and the high intensity of agricultural inputs [1]. For instance, current estimates indicate that an increase of 1 °C might cause a 10% to 20% reduction in the world's production of maize [2]. In fact, meta-analyses of climate change and its impact suggest that by 2030, crop yields may be decreased 50% [3].

The livelihood of millions of smallholder farmers and the survival of several national economies in Africa, Latin America, and Asia depend on crops usually considered commodities, or minor or orphan crops; examples are coffee (*Coffea arabica* L.) and cacao (*Theobroma cacao* L.) [4–6]. These crops are also believed to be at risk due to climate change [4,6]. For a tropical crop such as cacao, which is mostly grown in West Africa, the maximum temperature tolerated (38 °C) could be exceeded during hot and dry *El Niño* years, and the dry season may be extended for one additional month [6]. In Colombia, rising temperatures, longer droughts, and excessive rainfall have reduced coffee yields by 30% since 2008 [7]. Also for coffee, rapid deforestation and climate change may lead to the extinction of many wild African species and the permanent loss of diversity for future plant breeding [4].

Papaya is considered an important tropical crop because of its high nutritional value [8], especially its high content of vitamin A, vitamin C, thiamine, folate, niacin, riboflavin, iron, potassium and calcium [8]. Papaya is also cultivated for papain, an important proteolytic enzyme present in the latex of fruits harvested before ripening [9]. This enzyme improves digestion and can be used to cure ulcers [9]. Papain is also used for softening wool, preparing protein for animal food, the manufacture of cosmetics (toilet soap, toothpaste, and shampoo), and brewing beer [9]. Unfortunately, damage to photosynthetic carbon assimilation due to environmental stress, including water deficits, can reduce the biomass production and net carbon assimilation in papaya [9]. Therefore, a better understanding of the physiology and reproductive biology of papaya may facilitate its adaptation to climate change [9].

Unlike in major crops, minor tropical crops do not benefit from large public and private breeding programs that stress genomic and marker-assisted recurrent selection but rather focus on simple methods that prioritize backcrossing [5]. Hence, new approaches are urgently needed [5]. The following article outlines potential new approaches.

## 2. Fertility and Enhanced Meiotic Pairing

The development of new plant varieties by conventional plant breeding is based on the selection of traits already present in plant species. It relies heavily on sexual reproduction to accumulate favorable alleles for tolerance and resistance to stress, nutritional quality, or other agronomic and horticultural traits. Alleles that contribute to tolerance to stress or other traits can be obtained from local germplasm, landraces, breeding lines, wild species, or related genera [10]. *Theobroma cacao* has an anchored genome sequence of 324.7 megabase pairs (Mb) for the B97-61/B2 Criollo genome, which comprises approximately 28,798 predicted genes [11] (Table 1). The actual number of protein-coding genes is close to 21,437 [11]. About 24% of the genome consists of transposable elements such as the Gaucho long-terminal repeat (LTR) retrotransposon [12]. In cacao, the improvement of and selection for desirable traits is believed to have also caused accelerated accumulation of deleterious mutations because of population bottlenecks that started 3600 years ago [12,13]. Some of the mutations are suspected to be due to the process of fertilization itself [13]. However, as compared with non-domesticated *Theobroma cacao* varieties such as *Marañón* ($1.68 \times 10^{-5}$) and *Purús* ($1.23 \times 10^{-4}$), in the varieties *Amelonado*, *Contamana*, *Criollo* and *Guianna*, the same domestication process may account for differentially high recombination rates (expressed in centiMorgans per megabase pairs [cM/Mb], $4.04 \times 10^{-6}$ to $3.91 \times 10^{-3}$) [14]. Thus, selecting for high meiotic recombination rates during conventional breeding might help ameliorate the low individual fitness of modern varieties.

Little has been reported about the chromosomes and the genome of the 22 *Theobroma* species (Malvaceae) [15]. All *Theobroma* species have the same diploid chromosome number ($2n = 20$) and similar morphology and length (1–2 μm) [15]. Positive staining with DNA fluorescent stain chromomycin A3, which preferentially binds to G-C sequences, revealed the prophase chromosomes of *Theobroma cacao* and *T. grandiflorum* with two terminal heterochromatic bands that co-localize with a 45S rDNA site. Each chromosomal complement has a single 5S rDNA site in the proximal region of another chromosome pair [15]. Despite this apparent chromosome uniformity within the genus, meiotic analyses of a few cultivars of *T. cacao* have indicated the presence of univalents and multivalents, thus hinting at the existence of structural rearrangements in chromosomes [15].

An important consideration for breeding is that cacao has a long juvenile period, up to five years, so selection for fruit-related traits is expensive and time-consuming because the trees must be maintained for at least three more years in order to evaluate the pods visually [16]. Cacao tree plantations require a large area of land and labor force [16]. Because diseases are a persistent problem for cacao, improved disease resistance via breeding is imperative [17]. Each year, infection with a variety of pathogens severely cripples global cacao production, with about 30% of all pods destroyed before harvesting [17]. In West Africa, severe outbreaks of *Phytophthora* spp. can destroy all cacao pods on a farm [17]. The fungal pathogen *Moniliophthora perniciosa* causes Witches' broom disease, which seriously cripples cacao yields in Ecuador and Brazil [16]. Cacao is highly heterozygous and mostly involves outbreeding; thus, generating inbred lines from crosses is laborious, and doubled haploid lines are notoriously difficult to develop [16]. Nonetheless, there are a few self-compatible cacao clones, such as the Ecuadorean CCN51 [16] and Costa Rican CATIE-R1 (Figure 1A).

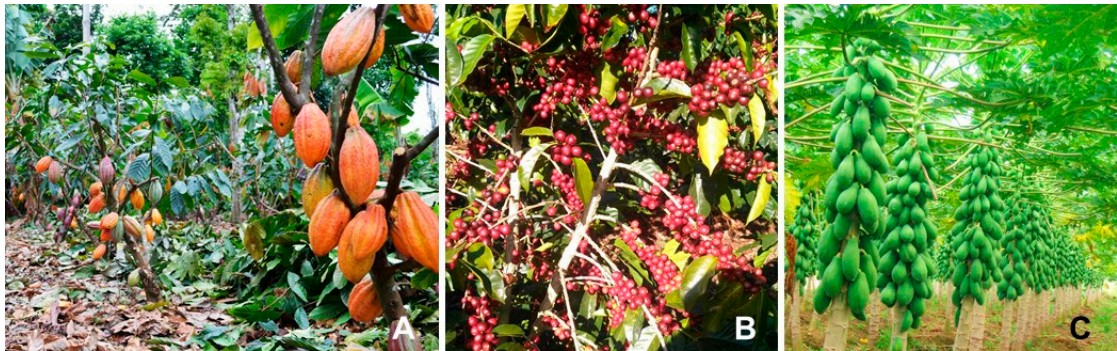

**Figure 1.** Tropical crops that may be amenable to genome editing and chromosome engineering, as actually grown in Costa Rica. (**A**) Cacao field plot of newly developed, self-compatible variety CATIE-R1 at CATIE, the Tropical Agricultural Research and Higher Education Center (Cartago province, Costa Rica). (**B**) Arabica coffee plants, Catuaí Yellow cultivar (León Cortés county, Costa Rica). (**C**) Papaya, Pococí hybrid (Alajuela province, Costa Rica). Credits, (**A**–**C**): Allan Mata-Quirós (CATIE, Costa Rica), Emmanuel López (Passiflora Coffee Farm, Costa Rica), Eric Mora–Newcomer (University of Costa Rica).

Cultivated coffee mostly corresponds to the species *Coffea canephora* Pierre ($2n$ = 22 chromosomes, 364 Mb) and *C. arabica* L. ($2n$ = 44 chromosomes), the latter likely due to hybridization between *C. canephora* and *C. eugenioides* [18]. The chromosome number for the genus *Coffea* is $n$ = 11, which is common for most genera of the family Rubiaceae [19]. Most *Coffea* species are diploid ($2n$ = 22) and self-incompatible; however, *C. arabica* L. is the only polyploid species ($2n$ = 44) of the genus, and it is self-compatible [19]. The genome of *C. canephora* is reported to be 569.9 Mb [20] (Table 1) and consists of 25,574 protein-coding genes, of which 23 are unique caffeine *N*-methyltransferases [18]. Approximately 50% of the genome consists of transposable elements, especially LTR retrotransposons of the *Copia* group [18]. Although they are morphologically different, *Coffea* species share common meiotic traits, such as little variation in the number of chiasmata and a high frequency of bivalents, even in interspecific hybrids [19]. Genomic in situ hybridization with total genomic DNA from *Coffea arabica* and *C. canephora* showed high cross-hybridization to chromosome preparations from *C. arabica*, *C. canephora*, and a *C. liberica*-introgressed *C. arabica* genotype [21]. All chromosomes can be labeled with these probes, which may suggest a close genetic relation between *Coffea arabica* and *C. canephora* and between the *Ca* genome of *C. canephora* and the *Ea* subgenome (from *C. eugenioides*), which are present in *C. arabica* [21].

Approximately 60% of the world's total coffee output is from *Coffea arabica*. The species is cultivated in Latin America, Ethiopia, Kenya, and Tanzania (see Figure 1B), whereas *Coffea canephora* is cultivated in Southern Brazil, Central Africa and Southeast Asia [22]. One of the key limitations for the breeding of new coffee varieties is the poor genetic diversity of current cultivars of *Coffea arabica*. This bottleneck is likely due to the limited number of individuals from the cultivars Típica and Bourbon that were introduced in the American tropics to initiate commercial plantations at the beginning of the 17th century [22].

It typically takes 25 years to develop a new coffee variety [22]. Caffeine is an alkaloid synthesized by several plants, including coffee and cacao [18]. Caffeine is synthesized in the leaves of coffee, where it may function as an insecticide, and in fruits and seeds, where it may inhibit the germination of competitors [18]. In the Arabica variety Caturra, famous for its good beverage quality, the caffeine content is approximately 1.15% (which is high); unfortunately, Caturra is susceptible to infection with coffee leaf rust (*Hemileia vastatrix*) and coffee berry disease (*Colletotrichum kahawae*) [22].

Papaya has a small genome, 372 Mb (Table 1); it is diploid ($2n$ = 18) and has two sex chromosomes, X and Yh [23], which are believed to have evolved about 7 million years ago [24]. Papaya pachytene chromosomal domains, brightly stained with fluorescent DNA stain 4′,6-diamidino-2-phenylindole

(DAPI), account for approximately 17% of the total papaya genome, which suggests that it is largely euchromatic [25]. The genome is believed to contain 21,784 protein-coding genes [23].

**Table 1.** Resources and methods for the study and transformation of genomes in cacao, coffee, and papaya.

| Species | Genome Database | Suggested Method for Engineering | Estimated Time for Engineering and Regeneration of Plants |
|---|---|---|---|
| *T. cacao* L., Criollo genotype (B97-61/B2) | The Cocoa Genome Hub, http://cocoa-genome-hub.southgreen.fr/ [11] | *Agrobacterium*-mediated transient transformation with CRISPR/Cas9 of cacao leaves and cotyledon cells [17]. | Five years for seedlings to reach sexual maturity [16]. Time required to perform engineering is unknown. |
| *C. canephora* P., accession DH200-94 | Coffee Genome Hub, http://coffee-genome.org [20] | Cocultivation of embryogenic calli of *Coffea canephora* with *Agrobacterium* [26]. | Six to 8 years to reach sexual maturity in *Coffea arabica* [27]. In *C. canephora*, induction of primary calli takes 1 month and induction of embryogenic calli takes an additional eight months. The selection of transformants takes at least one month, whereas plantlet regeneration may take another month. Total minimum estimated time: 11 months [26]. |
| *C. papaya*, 'Sun Up' | *Carica papaya* ASGPBv0.4, https://phytozome.jgi.doe.gov/pz/portal.html#!info?alias=Org_Cpapaya [28] | *Agrobacterium*-mediated transformation of somatic and zygotic embryos [29,30]. | Four months for plantlets to reach maturity [8]. Transformation takes a long time [31]. Following field cultivation of line Tainung #2 for one year, papaya shoots were collected and rooted in vitro with indole-3-butyric acid (time was not reported) [31]. Shoots were then cultured for six weeks to harvest adventitious roots [31]. Roots were cultured in vitro for three months to obtain somatic embryos [31]. Embryos were cocultured for two days and selected for 80 days [31]. Regenerants were cultured for one week to induce shooting and transferred to vermiculite in the greenhouse [31]. Estimated time: approximately one year and seven months. |

Papaya is a large herbaceous perennial plant with a short juvenile growth phase of only 4 months [8]. Once papaya reaches reproductive maturity, each leaf axil will develop flowers continuously [8]. It is a trioecious species, in that its flowers may develop into three sex configurations: female (X), male (Y or MSY) and hermaphrodite (Yh or HSY) [32]. Wild papaya features a 1:1 proportion of male to female plants (dioecious) and cultivated papaya a proportion of 2:3 hermaphrodite and 1:3 female plants (gynodioecious) [32]. Hermaphrodite flowers develop into elongated fruits with a small cavity (see Figure 1C), a trait that is sought-after commercially, as opposed to flowers from female plants, which are ovoid and have a large cavity [28,33]; thus, great effort has been placed on determining the sex of plants grown in the field. The X region is only 3.5 Mb, whereas the HSY and MSY regions are about 8.1 Mb and are located on chromosome 1 [32]. Notably, any combination of the HSY and MSY regions is inviable [32], which has been interpreted as meaning that the HSY and MSY regions may not carry the instructions required for embryo development [34]. HSY and MSY are extremely methylated and heterochromatic as compared with regions located on the X chromosome [34].

Transcriptome and whole-genome sequencing of papaya have revealed the existence of a putative gene called *SHORT VEGETATIVE PHASE (SVP)-LIKE*, which might be involved in male-hermaphrodite

flower differentiation and is present in naturally all-hermaphrodite populations of papaya grown in Pingtung, Taiwan [33]. The *SVP-like* gene is expressed specifically in papaya plants that are male (carry the MSY chromosome) and hermaphrodite (carry the HSY chromosome) but not in female plants (carry the X chromosome) [33]. The MSY chromosome appears to code for an intact SVP-like protein with both MADS- and K-box domains, the HSY chromosome codes for a partial K-box domain, and the X chromosome may code for a comparatively shorter fragment [33]. Close analysis suggests that in the HSY chromosome, the *SVP-like* gene has two insertions of *copia*-like retrotransposons [33], and a single polymorphism allows for identifying each insertion [33]. Other genes located on the HSY chromosome are also believed to mediate the development of hermaphrodite flowers, such as *CHROMATIN ASSEMBLY FACTOR 1 SUBUNIT A-LIKE* (*CpCAF1AL*) and *SOMATIC EMBRYOGENESIS RECEPTOR KINASE* (*CpSERK*) [34].

## 3. Mechanisms of Meiosis and Fertility

The process of meiosis has two purposes: to generate the genetic diversity transmitted by the gametes and to ensure correct segregation of chromosomes [35]. Defects in recombination and meiosis cause sterility [35]. In *Arabidopsis*, meiotic recombination begins with the formation of double-strand breaks (DSBs) in DNA, which is initiated by a topoisomerase-like protein called SPO11-1 and associated proteins such as MTOPVIB, PRD1–3, and DFO [36]. These breaks are processed by several proteins including MRE11, RAD50, NBS1 (the MRX complex) and Exo1 for endonucleolytic cleavage and removal of SPO11, COM1 and CtIP for end processing and by RPA1A-D and RPA2-3 for binding of single-strand DNA breaks at 3' ends and for recruiting recombinases [36]. Meiotic recombination results from the activity of RAD51, RAD51B-D, XRCC2, XRCC3, DMC1 and BRCA2 [36], among others.

Meiotic crossovers intermix homologous chromosomes to produce novel combinations of alleles that are transmitted to offspring [37]. The study of the formation of meiotic crossovers is important for activities related to plant breeding such as the detection and identification of quantitative trait loci and gene mapping [37]. Infertility results from defects in the formation of crossovers because the homologs segregate randomly during cell division [35]. Thus, at least one crossover must be formed per chromosome pair to ensure the formation of balanced and viable gametes [35].

Two major crossover formation pathways exist in plants. The major one depends on the ZMM protein complex (MSH4, MSH5, MER3, HEI10, ZIP4, SHOC1, PTD) plus MLH1/3 and produces interfering crossovers, whereas the minor pathway relies on MUS81 and produces non-interfering crossovers [36]. Crossovers are relatively rare events because active mechanisms limit their formation [37]. In the model plant *Arabidopsis*, three anti-crossover pathways rely on the activity of the proteins RECQ4, FANCM, and FIGL1 [37]. RECQ4 is the homologue of yeast Sgsp and is a DNA helicase; FANCM is also a DNA helicase (which requires cofactors MHF1-2), whereas FIGL1 encodes an AAA-ATPase [37]. In *Arabidopsis*, changes in their activity cause notable changes; for instance, recombination was increased four-fold in the *recq4a/recq4b* mutant but increased three-fold in the *fancm−/−* mutant [37]. The *recq4a/recq4b/figl* mutant showed an impressive eight-fold increase in recombination [37]. Work in peas, tomato and rice has shown that the identification of allele mutants for meiotic anti-helicases may boost recombination in crops to facilitate introgression of agriculturally valuable alleles [37].

Plants have two and possibly three homologous recombination pathways: (1) the standard DSB repair (DSBR) pathway and (2) the synthesis-dependent strand annealing (SDSA) pathway [38]. DSBR and SDSA pathways share common steps in inducing breaks but differ in how the resulting displacement loop (D-loop) is resolved [38]. In the standard DSBR pathway, strand exchange results in the initial establishment of a double Holliday junction (dHJ), and subsequent resolution of the dHJ leads to crossover formation between homologous chromosomes [38]. In the SDSA pathway, the dHJ is dissolved, which results in the formation of a non-crossover (gene conversion event) [38]. In somatic plant cells, homologous recombination of DSBs is believed to proceed through the non-crossover SDSA pathway [38]. Plants have another HDR pathway, single-strand annealing (SSA), that repairs DSBs

by using homologous sequences found within the same DNA sequence [38]. In the SSA pathway, an adjoining region of complementarity is used as the annealing site for the free ends of the break, and ends that are non-complementary are trimmed [38]. Thus, the SSA pathway leads to the loss of sequences located between the two repeat regions [38]. The FANCM helicase is believed to be involved in both the SDSA and SSA pathways [38], and its cofactors MHF1/2 are also considered to have a similar role in limiting crossovers and to act genetically in the same pathway [36] (see Figure 2A).

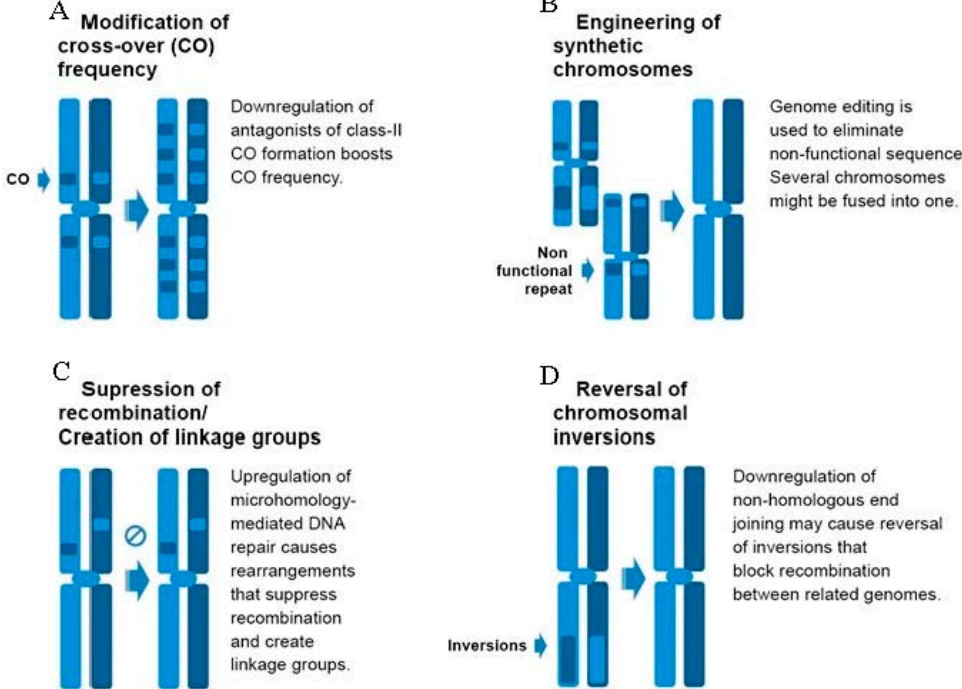

**Figure 2.** Possibilities for genome and chromosome engineering in tropical crops. (**A**) Modification of cross-over (CO) frequency, elimination or downregulation of meiotic antagonists of class II cross-over formation may lead to a significant increase in non-interfering cross-over formation, which may facilitate introgression of beneficial alleles from wild relatives. (**B**) Engineering of synthetic chromosomes, following the example of synthetic yeast strains, genome editing in plants may allow for the trimming of non-functional sequences, such as those corresponding to transposable elements, and facilitate the aggregation of open reading frames into so-called "Megachunks". Eventually, megachunks from different chromosomes may be combined into few, fully functional chromosomes. (**C**) Suppression of recombination and creation of linkage groups. In somatic tissues, downregulation of non-homologous end joining (NHEJ) by targeting either the Ku70/Ku80 heterodimer or the XRCC4 ligase leads to a shift to repair through backup EJ by DNA polymerase θ, which results in large deletions, inversions and translocations. These rearrangements may block recombination and lead to the de novo creation of new linkage groups that facilitate apomictic-like asexual reproduction. This trait may be useful for harvesting hybrid seeds that breed true to type in the next generation. (**D**) Downregulation of Ku70/Ku80 may lead to large chromosomal inversions up to 18 Kb in length, which may help undo chromosomal rearrangements that may occur during the evolution of new closely related species; such ancestral rearrangements may result in difficult recombination between chromosomes from commercial varieties and those from wild relatives. Downregulation of NHEJ may also facilitate CRISPR/Cas9-mediated genome editing.

Besides using homologous repair, plants may repair DSBs in their DNA by non-homologous end joining (NHEJ) via the Ku-dependent, "standard" NHEJ pathway or the highly erring Ku-independent backup-EJ pathway, especially in somatic cells, which are often the tissue of choice for genetic engineering experiments [38]. The backup-EJ pathway is also called microhomology-mediated end joining [38]. In canonical NHEJ, the Ku70-Ku80 heterodimer and the XRCC4 ligase recognize DSBs

and bind to them, with minimal end processing, which results in minimal DNA loss (1 to 4 nt) [38]. In backup EJ, DSBs are bound by poly (ADP-ribose) polymerase proteins and the MRX complex in order to promote end-resection and generate homology between the two DNA sequences; however, the broken DNA ends are resected and extended by the very error-prone DNA polymerase θ, which results in large deletions, inversions, and translocations [38]. Downregulation of NHEJ genes is believed to shift most DNA repair toward homologous recombination and thus increase the efficiency of CRISPR/Cas9 editing [39,40].

Other possibilities for chromosome engineering have been suggested for *S. cerevisae* [41] and could be considered for engineering synthetic chromosomes in plants. For instance, all sub-telomeric repeats and retrotransposons could be removed until chromosomes can be organized into regular megachunks composed of fully functional open reading frames [41]. Methods could be developed to control the chromatin state of entire chromosomes [42] and even to fuse all chromosomes into only one as done in yeast with CRISPR/Cas9 [43] (see Figure 2B).

## 4. Possibilities for Genome Editing in Coffee, Cacao and Papaya

Genome editing is a technological breakthrough that allows for the creation of targeted mutations and the replacement of sequences with high specificity [26]. The CRISPR/Cas9 editing technology is inspired by bacterial adaptive immune systems that rely on clustered regularly interspaced, short palindromic repeats (CRISPR)-associated protein 9 (Cas9) whose function is to protect against invading foreign DNA by cleaving it with an RNA-guide [26]. The system is simple and efficient [26] and requires a trans-activating crRNA sequence (tracrRNA) to assist in the maturation of CRISPR RNAs (crRNAs) [44].

The most basic gene editing process by CRISPR relies on the formation of DSBs and appears to have 2 stages [45]. The first stage is recognition of a palindrome (PAM) [45,46]. This stage is determined by the sequence of PAM itself, by interactions with Cas9 interactions and by chromatin accessibility [45]. The second stage is the formation of the R-loop, which is a structure between the target DNA (taDNA) and the signal guide RNA (sgRNA) [45]. Usually, the formation of an RNA-DNA hybrid helix 20 bp long is required for sgRNA-target recognition and Cas9 cleavage, but shorter sequences (protospacers) may also work. The Cas9 endonuclease may tolerate mismatches in the distal region of the PAM (non-seed region), although perfect base-pairing in the proximal region of the PAM-proximal region (seed region) is preferred [45]. The PAM motif that Cas9 may recognize is 5′-NGG-3′ (where N = A, T, C, or G), but unfortunately, this G-rich sequence makes it difficult to design sgRNAs located within T-rich regions [44]. An alternative is to resort to Cpf1, a class 2 endonuclease in the CRISPR system, that is efficient in plant genome editing. Quite conveniently, the Cpf1 endonuclease does not require a tracrRNA sequence to form a mature crRNA, and it can also recognize T-rich PAM sequences. Additionally, cutting by the Cpf1 endonuclease produces cohesive ends, unlike Cas9, which produces blunt ends [44]. In addition to generating insertions and deletions at target sequences, CRISPR/Cas systems have been developed for base editing [44]. These base editors consist of a sgRNA-guided Cas9 nickase (e.g., nCas9) fused with a deaminase that causes very precise C-T or A-G base conversions [44].

In contrast to animal cells, plant zygotic cells are difficult to transform with CRISPR/Cas9 directly due to technological limitations [46]. Currently, the transformation of most plant species involves callus cells obtained by tissue culture and exposure of those cells to *Agrobacterium* [46]. The efficiency of transformation may depend on the choice of codons for Cas9 translation and the promoters used for the expression of Cas9 and the sgRNAs [46]. An exciting recent development is prime editing, a "search-and-replace" technology that facilitates targeted insertions, deletions, and base-to-base conversions without the need for DSBs [47]. This is possible because of prime editors (heavily modified Cas9 proteins) that feature a reverse transcriptase (RT) fused to an RNA-programmable nickase and a prime-editing guide RNA (pegRNA), all working together to copy genetic information from the pegRNA into the target sequence [47].

For cacao, cells isolated from somatic embryo cotyledons and young leaves (Scavina 6 variety) can be transiently transformed by using *Agrobacterium* carrying a CRISPR/Cas9 construct [17] (Table 1). Apparently, stable expression was achieved with the vector pGSh16.1010, which carries the Cauliflower Mosaic Virus (CaMV) *35S* promoter [17]. Coffee has been successfully transformed by using embryogenic callus cultures as the tissue of choice for transformation with *Agrobacterium tumefaciens* (strain LBA1119) [48] (Table 1). For the *Coffea canephora* clone 197, the *C. canephora U6* promoter was used to transform 6-month-old embryogenic calli derived from leaf segments for 10 min with a solution containing the *Agrobacterium tumefaciens* strain EHA105, selected for tolerance to cefotaxime and hygromycin and grown into plants [26]. A pCAMBIA 5300 binary vector carrying the Cas9 sequence was successfully used for these transformants [26]. Approximately 7% of all transformants tested homozygous for mutations of the phytoene desaturase gene (*CcPDS*, *Cc04_ g00540*) [26].

For papaya, genetic engineering has mostly involved biolistic-mediated transformation related to tolerance to the papaya ringspot virus, a Potyvirus that is quite possibly the most serious viral disease of papaya worldwide [29,30]. Nonetheless, *Agrobacterium*-mediated transformation of somatic and zygotic embryos has also been reported [30] (Table 1). CRISPR/Cas9-mediated genome editing against several viruses simultaneously in papaya may be technically feasible and within reach [30].

## 5. Chromosome Structure Editing in Crops

Besides the targeted editing of key genes, genome engineering may also allow for the modification of chromosome structure [40]. Theoretically, if several DSBs in DNA are introduced into the genome, chromosomes might be modified in a directed manner to create new combinations of fragments [40]. For instance, if two DSBs are induced on the same chromosome, deletions or inversions in the area in between may give rise to intrachromosomal rearrangements [40]. However, to create interchromosomal rearrangements, the formation of two or more DSBs on different chromosomes would be required [40]. Depending on the type of tissue, a somatic or meiotic crossover may be formed by inducing breaks on one or both homologues [40].

In *Arabidopsis*, inversions of about 18 kb in length were successfully passed on to offspring by using an egg cell-specific promoter (*EC1.1*) for Cas9 (SaCas9) expression [49]. Inversion and deletion frequencies were reported to be greater in a *ku70-1* mutant than the wild type, which suggests increased efficiency by mutagenic microhomology-mediated backup EJ [49]. This process has been observed in mice as well [50]. In agriculture, this approach may help reverse natural inversions between related species to facilitate outcrossing and transmission of beneficial alleles [49,50] (see Figure 2C). For example, in papaya, crosses of species from the related genus *Vasconcella* might promote transmission of tolerance to ringspot virus and the fungus *Phytophthora palmivora*, which causes fruit rot and destroys the root meristems [51]. However, meiosis is usually defective in these wild species [51], and lagging chromosomes and polyads are observed [51].

Another yet theoretical possibility for crops would be to induce an inversion to create new linkage groups and purposely suppress recombination [40] (see Figure 2D). Suppression of meiotic recombination is also observed during apomixis, a form of reproduction characterized by deregulation of meiosis during embryo sac formation [35], development of the embryo in a fertilization-independent way [35], and formation of viable endosperm in a fertilization-dependent or -independent way [35]. Apomixis is an effective strategy to retain genome-wide parental heterozygosity; recently, apomixis was successfully engineered in rice to facilitate clonal propagation of hybrids by seed [52].

## 6. Concluding Remarks

Advances in understanding agricultural genomes in tropical crops such as coffee, cacao, and papaya may hopefully allow for adaptation to climate change and increased output. It is important to take timely advantage of the possibilities brought about by genome editing technologies in order to boost recombination and crossover formation and as a result, improve the transmission of novel traits. It may also be possible for the first time to take advantage of the downregulation of NHEJ to

create novel linkage groups and even to reverse natural chromosomal inversions between related species. Farmers in developing countries and their customers in developed economies may all benefit immensely from the adoption of these technologies.

**Funding:** This research was funded by Vicerrectoría de Investigación (University of Costa Rica) grant numbers B8069, B6602 and B5A52. And the APC was funded directly by Vicerrectoría de Investigación.

**Acknowledgments:** Work at the Bolaños-Villegas laboratory was supported by grants B8069, B6602, B5A52 and B5A49 from Vicerrectoría de Investigación (University of Costa Rica), which also covered publication costs for this article. Pablo is a young member affiliate of TWAS/UNESCO and a member of the American Society of Plant Biologists. This manuscript was kindly edited by Ms. Laura Smales (BioMedEditing, Toronto, Canada). Images were kindly edited by E. Bolaños-Villegas.

**Conflicts of Interest:** The author declares no conflicts of interest. The sponsors had no role in the design, execution, interpretation, or writing of the study.

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
