# Peer review of "Chromosome Engineering in Tropical Cash Crops"

_agronomy, doi:10.3390/agronomy10010122_

Round 1

Reviewer 1 Report

I think that the author has not addressed with sufficient detail my previous comments. In addition, I have not seen a point-by-point response to them.

Author Response

Dear reviewer. I am sorry I did not address your observations properly. I will look into your previous comments to look for each one and how it was handled. Nonetheless I will appreciate greatly if you could tell me, in general terms, what do you think are the most obvious weaknesses of the manuscript.

Reviewer 2 Report

Minor changes

1) Correct symbol for degrees – such as line 24, 38 etc.

2) Confirm that the statement on line 71 is correct (0.5 and 2.0 mm) – in the sitation #15 I found this info in the abstract  “A comparative analysis of mitotic chromosomes of Theobroma cacao (cacao) and T. grandiflorum (cupuaçu) was performed aiming to identify cytological differences between the two most important species of this genus. Both species have symmetric karyotypes, with 2n = 20 metacentric chromosomes ranging in size from 2.00 to 1.19 µm (cacao) and from 2.21 to 1.15 mm (cupuaçu)“

3) Many terms should be explained better. 

For example:

Line 66 – cM/Mb – these could be explained when they are mentioned first time

Line 72 - chromomycin A3 – what does it specifically stain? 

Line 127 “sex chromosome system” – not explained at all.

Overall there are a lot of concepts that are cited but not explained. Since this is a review article it should be all explained properly.

Author Response

Dear reviewer. Thank you for your comments, I will check the manuscript to address each comment. Would you kindly tell me which other concepts are not properly explained?. 

Reviewer 3 Report

Dear Pablo,

You have included most requested corrections. I am satisfied with the present manuscript and consider that it should be accepted as it is now. Congratulations!

Author Response

Thank you. You are very kind. I appreciate your kind comments and encouragement. 

Best regards,

Pablo

Round 2

Reviewer 1 Report

Comments to the author: - "I must apologize. My skills on image design are poor” is not an appropriate answer. I think that an additional figure is required. - If the focus of the paper is on crop genetic engineering, why do you insist in a section about microbial metabolic engineering? If the focus were on the production of biocompounds of interest I could understand a section for plant-based production and other for bug-based production. - The conclusion needs more punch, otherwise we can skip the reading of the paper.

Author Response

Dear reviewer,

Hello. I also felt a figure on engineering was needed. I added one, albeit a simple one. I also removed the part on metabolite engineering. It was too convoluted. The conclusion was also rewritten. 

Thanks for your valuable time and frank comments.

Best regards,

Pablo